# Melatonin Prevents Alcohol- and Metabolic Dysfunction- Associated Steatotic Liver Disease by Mitigating Gut Dysbiosis, Intestinal Barrier Dysfunction, and Endotoxemia

**DOI:** 10.3390/antiox13010043

**Published:** 2023-12-25

**Authors:** Karli R. LeFort, Wiramon Rungratanawanich, Byoung-Joon Song

**Affiliations:** Section of Molecular Pharmacology and Toxicology, National Institute on Alcohol Abuse and Alcoholism, 9000 Rockville Pike, Bethesda, MD 20892, USA; wiramon.rungratanawanich@nih.gov

**Keywords:** melatonin, alcohol-associated liver disease, metabolic dysfunction-associated steatotic liver disease, intestinal barrier dysfunction, Cyp2e1, oxidative stress, antioxidant, post-translational modifications, mitochondrial dysfunction, gut–liver axis

## Abstract

Melatonin (MT) has often been used to support good sleep quality, especially during the COVID-19 pandemic, as many have suffered from stress-related disrupted sleep patterns. It is less known that MT is an antioxidant, anti-inflammatory compound, and modulator of gut barrier dysfunction, which plays a significant role in many disease states. Furthermore, MT is produced at 400–500 times greater concentrations in intestinal enterochromaffin cells, supporting the role of MT in maintaining the functions of the intestines and gut–organ axes. Given this information, the focus of this article is to review the functions of MT and the molecular mechanisms by which it prevents alcohol-associated liver disease (ALD) and metabolic dysfunction-associated steatotic liver disease (MASLD), including its metabolism and interactions with mitochondria to exert its antioxidant and anti-inflammatory activities in the gut–liver axis. We detail various mechanisms by which MT acts as an antioxidant, anti-inflammatory compound, and modulator of intestinal barrier function to prevent the progression of ALD and MASLD via the gut–liver axis, with a focus on how these conditions are modeled in animal studies. Using the mechanisms of MT prevention and animal studies described, we suggest behavioral modifications and several exogenous sources of MT, including food and supplements. Further clinical research should be performed to develop the field of MT in preventing the progression of liver diseases via the gut–liver axis, so we mention a few considerations regarding MT supplementation in the context of clinical trials in order to advance this field of research.

## 1. Introduction: MT

Melatonin (MT), known as *N*-acetyl-5-methoxytryptamine, is a sleep-regulating hormone known to be secreted from the hypothalamus–pineal gland axis [1]. However, recent reports demonstrate that the gastrointestinal tract (GIT) is responsible for producing 400 [2] to 500 [3] times more MT than the pineal gland, suggesting that MT may be more prominent in regulating the gut and other peripheral organs [3,4,5].

This review details how MT made from the intestines interacts in the gut–liver axis to prevent liver diseases. We have explained that MT produced in intestinal mitochondria targets hepatic mitochondria by regulating oxidative stress mainly generated by one-electron leakage from the electron transport chain [6] as well as from the activities of cytochrome P450-2E1 (CYP2E1), inducible nitric oxide synthetase (iNOS), and NADPH-dependent oxidases (NOXs). Consequently, MT attenuates mitochondrial dysfunction, and mitigates metabolic alterations and inflammatory processes that are involved in the progression of liver diseases. For instance, MT prevents the worsened progression of alcohol-associated liver disease (ALD) and metabolic dysfunction-associated steatotic liver disease (MASLD) by protecting the intestines against gut dysbiosis, intestinal barrier dysfunction, and endotoxemia.

### 1.1. Synthesis of MT

MT synthesis mainly occurs in the brain and intestines. In the brain, MT synthesis occurs in pinealocytes, which are photoreceptor cells [7,8,9]. Intestinal MT synthesis occurs in enterochromaffin cells in the mucosal layer of the villi [10,11]. MT production in the brain and gut is highly influenced by environmental stimuli such as the light–dark cycle or dietary exposures to the precursor of MT, L-tryptophan [7,8,9]. It is theorized that intestinal MT synthesis is stimulated mainly by feeding cues, and photoreceptor synthesis of MT (stimulated by light–dark cycles) occurs within an opposite phase, suggesting that MT is constantly being synthesized [11]. MT production in the gut may also be inversely related to cortisol levels, as these two hormones are involved in circadian rhythm-coordinated metabolism in the brain and many peripheral tissues [1,5,12].

In all cells that synthesize MT, L-tryptophan is hydroxylated to 5-hydroxytryptophan in mitochondria and then converted to 5-hydroxytryptamine (serotonin) in the cytoplasm [13]. Serotonin is then acetylated to *N*-acetylserotonin and converted to MT [13]. In total, four enzymes including tryptophan-5-hydroxylase (TPH), L-aromatic amino acid decarboxylase (AADC), arylalkylamine *N*-acetyltransferase (AA-NAT), and *N*-acetylserotonin-*O*-methyltransferase (ASMT), as well as three membrane receptors including MT1, MT2, and MT3 are involved in MT production and signaling [14].

While MT supplementation is widely known, sufficient amounts of MT can be consumed in foods high in its precursor, L-tryptophan, and in organisms associated with food cultivation. MT can be produced from many types of bacteria, protists, fungi, macroalgae, plants, and animals and is abundant in fruits, vegetables, nuts, seeds, grains, and olive oil (see Section 5 for more sources) [15,16,17,18]. In general, the Mediterranean Diet also contains MT, but it should be noted that foods have certain health benefits via the synergistic effects of the combination of compounds in their composition [12]. MT may even be produced by human commensal bacteria, as reviewed [3,4,5,19,20].

MT stores are distributed throughout the body in the intestines, brain, eyes, skin, bones, blood, immune system, thymus, spleen, GIT, inner ear, pancreas, liver, kidneys, reproductive tract, upper respiratory tract, heart, and thyroid [5]. MT is also found in many bodily fluids, including blood, amniotic fluid, synovial fluid, bile, saliva, breast milk, and cerebrospinal fluid (CSF) [12]. Importantly, it should be noted that MT accumulates in these tissues and contributes to the circadian rhythm of all these tissues/fluids. Likewise, the metabolic functions of these tissues are interrelated and are affected by depleted MT stores. MT has been shown to regulate the functions of the immune system, heart, gut, liver, kidney, reproductive organs, and endocrine glands and to act in organ axes to prevent chronic and acute conditions [21,22].

Given the number of cells in which MT is stored and synthesized, it is essential to support sufficient biological stores of MT in our cells for optimal homeostatic communication and prevention of liver diseases. In sum, this review details how supplementation and restored levels of MT affect several components in the gut–liver axis to prevent the progression of ALD and MASLD.

### 1.2. Contributing Factors of ALD within the Gut–Liver Axis

The liver is the main site of oxidative and non-oxidative ethanol metabolism, including enzymes such as alcohol dehydrogenase, aldehyde dehydrogenase 2 (ALDH2), Cytochrome P450-2E1 (CYP2E1), and iNOS [23,24]. Redox imbalance [25], poor nutritional status [26], mitochondrial dysfunction [27], cellular injury (apoptosis/necrosis) [28], inflammation [29], altered cell signaling pathways [30], and endotoxemia [31] all contribute to the pathology of ALD [25,26,27,28,29,30,31].

Fat accumulation (steatosis) can happen from increased *de novo* lipid synthesis, lipid transport from adipose tissues, and mitochondrial dysfunction that causes decreased fat degradation [32]. However, alcohol-induced accumulation of reactive oxidative species (ROS) [33], lipid peroxides (LPOs) (e.g., acrolein, malondialdehyde (MDA), and 4-hydroxynonenal (4-HNE)) [34], and hepatocyte death pathways may all contribute to the activation of hepatic stellate cells (HSCs), which then stimulate fibrosis and cirrhosis, leading to hepatocellular carcinoma (HCC) or liver failure [32]. One study demonstrated that ROS-, LPO- and hepatocyte apoptosis-mediated HSC activation may take place via the increased production and accumulation of α-smooth muscle actin (α-SMA) and collagen in the extracellular matrices, upregulated pro-fibrogenic metalloproteinase-2 (MMP2), and downregulated anti-fibrogenic metalloproteinase-1 (MMP1) via the MDA/4-HNE pathway [25,35,36].

Chronic and binge alcohol exposure can cause intestinal barrier dysfunction, worsening the progression of liver disease [37,38,39,40]. Elevated intestinal permeability makes endotoxemia a likely outcome, measured by increased serum LPS [25,41]. Acetaldehyde, a highly reactive aldehyde produced during oxidative ethanol metabolism, can increase permeability in the intestinal barrier by disrupting the distribution and organization of tight junction and adherens junction (TJ/AJ) proteins in the intestines [25]. Alcohol (ethanol) also changes the composition of the microbiome in ALD [31,42,43,44,45], even at the early stages of ALD, causing dysbiosis and bacterial translocation [45]. Even AUD patients exhibit microbial characteristics such as upregulation of GABA metabolism pathways that could serve as intestinal fingerprint biomarkers with a 93% accuracy [29,46].

Alterations of ALDH2, CYP2E1, and iNOS can also contribute to intestinal permeability and endotoxemia in the progression of ALD. Inhibition of ALDH2 can directly increase intestinal permeability via elevated acetaldehyde, as demonstrated in *Aldh2*-KO mice exposed to the alcohol liquid diet for four weeks [47] or binge alcohol [48]. LPS released from the gut interacts with TLR4 in specialized macrophages in the liver (Kupffer cells) and other immune cells to stimulate an inflammatory response and ROS production [49]. These changes promote hepatocyte damage, activate hepatic stellate cells, and contribute to fibrosis [49].

Another cellular mechanism of hepatic inflammation via the gut–liver axis may happen when LPS-mediated induction of iNOS results in toxic peroxynitrite production in the presence of ROS [23,24]. Elevated peroxynitrite can nitrate or *S*-nitrosylate proteins and promote the accumulation of 4-HNE protein adducts in mitochondria [50,51]; these post-translational modifications (PTMs) of various proteins in many subcellular compartments, including mitochondria, stimulate mitochondrial dysfunction and caspase-mediated apoptosis of hepatocytes [50,51].

Lastly, CYP2E1 is a significant contributor to alcohol-induced oxidative stress and ALD in periods of excessive alcohol use [36,52,53,54,55,56]. CYP2E1 is expressed in the intestine, induced by alcohol exposure, and involved in oxidative intestinal injury [57,58]. Specifically, it is induced by protein stabilization (i.e., protection from its rapid degradation via the ubiquitin-proteosome-dependent proteolytic pathway) after alcohol exposure [59,60]. CYP2E1-mediated ROS can also stimulate endotoxemia and elevate serum levels of LPS, which can travel to the liver, upregulating pro-inflammatory mediators, including tumor necrosis factor-alpha (TNF-α) [61] and hypoxia-inducible factor 1-alpha (HIF1-α) [49]. Elevated TNF-α and HIF1-α can activate mitogen-activated protein kinases (MAPKs) and decrease autophagy [49,61]. Upregulated CYP2E1 activity can also promote various PTMs, such as oxidation, S-nitrosylation, nitration, MAPK-mediated phosphorylation, acetylation, and adduct formations in mitochondria and endoplasmic reticuli, leading to mitochondrial dysfunction, endoplasmic reticulum (ER) stress, and apoptosis of hepatocytes and other cells like gut enterocytes [24,62]. CYP2E1-mediated oxidative stress can also sensitize the liver to toxicity via endotoxemia and intestinal TNF-α via a CYP2E1-thioredoxin-ASK1-JNK1 pathway [36,52,53,54,55,56,57].

### 1.3. Contributing Factors of MASLD via the Gut–Liver Axis

The incidence of obesity and metabolic syndrome has steadily increased over the last two decades. Obesity is affiliated with insulin resistance and metabolic syndrome, which may negatively affect the function of the liver, in addition to the heart, blood vessels, kidneys, and muscle mass [63]. MASLD and metabolic dysfunction-associated steatohepatitis (MASH) are considered co-morbidities of obesity and Type II Diabetes Mellitus that may be caused by substances other than alcohol, such as fructose/sucrose and Western-style high-fat diets (HFDs) containing high levels of n-6 fatty acids [64,65]. MASLD may also have hepatic manifestations similar to ALD. Like the progression of steatosis in ALD, increased de novo fat synthesis, fat transport from adipose tissues, and decreased fat degradation are typically seen in MASLD/MASH [66]. Oxidative and nitrative stress can significantly contribute to the progression of MASLD/MASH, partly via the involvement of CYP2E1 [30,67,68,69], additional oxidative stress [70], and various PTMs [30,62,67,68,69,70]. Oxidative stress and PTM accumulation can ultimately lead to mitochondrial dysfunction with decreased lipid degradation and insufficient energy (ATP) supplies [71]. CYP2E1 has been found to contribute to liver injuries caused by many substances including high fructose [64], HFDs [72], carcinogens [73], narcotics [74], pharmacotherapies [75], acetaminophen (APAP) [76,77], hepatotoxic substances to model MASLD (i.e., carbon tetrachloride (CCl_4_) [78,79] and thioacetamide (TAA) [78]), and nicotine [80]. CYP2E1 can also play a causal role in creating oxidative stress in MASLD [30,68,72,81] and insulin resistance [52,81,82,83].

It is well established that multiple hits are needed to initiate and develop MASLD/MASH [84]. MASLD has a wide variety of appearances, from as little as a simple accumulation of lipid droplets to active inflammation seen in MASH, with a small part progressing to fibrosis and HCC, and during this progression, multiple events occur. The original report of the “Two-hit Hypothesis” [84] described the development of steatosis as the first hit, and increased oxidative stress represents part of the second hit in the progression of MASLD [84]. According to others, the second hits also include intestinal barrier dysfunction [46], endotoxemia [46], inflammation [46], changed hepatocyte apoptosis pathways, and stimulated HSCs, all of which allow the evolution of milder cases of MASLD to progress to more severe cases, including MASH and fibrosis [46,66,85,86]. Others build on this hypothesis by suggesting that the “Multiple-hit Hypothesis” involves dysregulated lipid metabolism, mitochondrial dysfunction with suppressed fat degradation, and oxidative stress [66,85,86]. However, it is still unclear whether MASLD stimulates mitochondrial dysfunction and oxidative stress or if mitochondrial dysfunction and oxidative stress cause MASLD [66,85,86].

An explanation for this phenomenon is that chronic inflammation triggers oxidative stress, which stimulates more inflammation in a vicious cycle during the progression of MASLD [68,81,87,88]. During the phase of steatosis, mitochondrial respiration is increased to meet the elevated need for energy [89]. Mitochondrial respiration will increase the production of ROS and activate antioxidant responses [89]. Lipid accumulation may manifest as an excess of free fatty acids (FFAs), which can activate stress-activated *c*-Jun protein kinase JNK, leading to apoptosis of hepatocytes accompanied by damaged mitophagy of mitochondria [89]. Increased inflammation and oxidative stress trigger apoptosis, mitochondrial respiration, mitophagy, and antioxidant pathways that may compensate depending on the conditions. During cirrhosis, few compensatory mechanisms exist, suggesting that there are low levels of respiration, biogenesis, antioxidant pathways, or mitophagy. Further stimulation of nonparenchymal cells, such as HSCs, as well as inflammation, oxidative stress, and increased apoptosis of hepatocytes, occur at later stages of liver diseases [90,91].

Gut dysbiosis and endotoxemia are certainly causal risk factors in MASLD/MASH [28,92] and other liver disorders [93,94], and the intestinal “fingerprint” caused by the susceptibility and progression of MASLD differs from that of ALD [94]. For example, one study found that while patients with ALD tend to have a decreased abundance of Ruminococcus and *Faecalibacterium prausnitzii*, patients with MASLD had an increased abundance of Proteobacteria, Enterobacteriaceae, and Escherichia [94]. Furthermore, microbial products or metabolites such as LPS, short-chain fatty acids (SCFAs), bile acids, choline, trimethylamine-N-oxide (TMAO), and endogenously produced ethanol in the gastrointestinal tract [95,96,97] have been found to accumulate in the progression of MASLD [64,95,96,97,98]. More importantly, the intestines contribute to the progression of MASLD via excessive oxidative stress and systemic inflammation from leaked LPS [99]. The intestines may even alter lipid metabolism pathways via microbial SCFA metabolites [100] and disrupt mitochondrial function, leading to hepatocyte death pathways, inflammation, and fibrosis [101].

### 1.4. Functions of MT in the Gut–Liver Axis

The liver and gut contain cytochrome P450 isoforms involved in the deacetylation of *N*-acetylserotonin to form MT and the subsequent formation of 6-hydroxy-melatonin [102]. The pineal gland is responsible for the hydroxylation and decarboxylation of tryptophan to MT. However, the liver and intestine are among the few organs that express AA-NAT and ASMT enzymes to synthesize MT from serotonin [103]. AA-NAT and ASMT handle acetylation and then methylation of serotonin molecules to MT, respectively [3,5,104]. MT is also metabolized by cytochrome P450 isoforms in the liver when ingested orally or administered intraperitoneally [5]. Given its metabolism and localization in the liver and intestines, MT is likely to exert its beneficial effects on the gut–liver axis.

The metabolism of MT contributes to its unique function to combat oxidative stress in the gut-mitochondria axis. To begin, MT is a unique antioxidant because it produces metabolites that scavenge free radicals as well. MT can donate electrons to two free radicals or be hydroxylated to produce *N*-acetyl-*N*-formyl-5-methoxykynuramine (AFMK), which is then oxidized to another metabolite called *N*-acetyl-5-methoxykynuramine (AMK) [105,106]. AMK has been demonstrated to have many effects on oxidative stress in mitochondrial metabolism, including inhibiting and downregulating cyclooxygenases and suppressing iNOS and mitochondrial nitric oxide synthase (mtNOS) enzymes [106]. MT also directly upregulates the electron transport chain, oxidative phosphorylation, and ATP synthesis in mitochondria by interacting with complexes I and IV in healthy cells. In addition, MT has been shown to recover glutathione levels and decrease the activity and expression of iNOS and mtNOS in models of septic damage to mitochondria [107,108]. Thus, MT has an exponential antioxidant effect on mitochondrial metabolism by directly scavenging free radicals and producing metabolites that also scavenge free radicals.

#### 1.4.1. Intestinal Barrier Dysfunction

The function of MT in intestinal health may be revealed via studies on the effects of intestinal MT deficits in sleep-deprived clinical and animal models. MT deficits have been reported in night shift workers showing signs of intestinal barrier dysfunction and increased sensitivity to oxidative stress, inflammation, and several diseases, including ALD, MASLD, ischemic liver injury, and other metabolic disorders that may relate to intestinal barrier dysfunction [109,110]. It has been well documented that having a deficit of MT can result in multi-organ damage, most notably those tied to the gut [41]. These models of disrupted circadian rhythms may be correlated with elevated gut dysbiosis and intestinal barrier dysfunction [111,112]. A clinical study involving people with alcohol use disorder (AUD) reported decreased hours of sleep and MT levels, as well as increased intestinal hyperpermeability and endotoxemia [113], as recently reviewed [114]. Based on these studies, MT deficits are associated with alcohol-induced susceptibility to intestinal barrier dysfunction.

#### 1.4.2. Oxidative Stress and PTMs in Mitochondria

MT is primarily localized in the mitochondria of many cells [115,116,117,118], maintaining energy homeostasis [116,119,120] and reducing oxidative stress. Reactive oxygen species (ROS) are mainly produced from the one-electron leakage from the electron transport chain [6]. ROS may also be produced by the ethanol-inducible CYP2E1 enzyme [62,121], which is typically localized in microsomes but may translocate to mitochondria [122,123,124], causing mitochondrial dysfunction and fat accumulation [62,125]. MT also counteracts oxidative stress because it improves the functions of mitochondria in hepatocytes [126]. MT receptors, including MT1, MT2, and MT3, are localized in the gut, and MT3, also known as quinone reductase 2, has been known to have antioxidant properties [127]. MT also up-regulates NAD^+^-dependent protein-deacetylase Sirtuins (SIRTs) such as Sirtuin-1 (SIRT1) [128,129,130,131,132], Sirtuin-3 (SIRT3) [117,118,131,132,133,134,135,136,137], and Sirtuin-6 (SIRT6) [138,139], as shown in Figure 1. 

Sirtuins prevent the accumulation of protein acetylation, one type of PTM, in mitochondria, preventing mitochondrial dysfunction in liver diseases and aging [140]. MT also promotes mitophagy and autophagy in cells [134,135,137,141,142,143,144,145] and reduces apoptosis [146,147], protecting various organs from acute tissue injury and aging-related chronic diseases. In later sections, we will discuss more specific mechanisms by which MT exerts its effects as an antioxidant to prevent chronic and acute liver injuries via the gut–liver axis [128,148,149].

#### 1.4.3. Gut–Mitochondria Axis

Changes in the intestinal microbiota (i.e., gut dysbiosis) can lead to increased intestinal permeability and local inflammation, which can cause endotoxemia and decrease the production of SCFAs [150,151]. A high abundance of Gram-negative bacteria typically contributes to increased LPS and lowered SCFA production, which is commonly seen in intestinal barrier dysfunction and endotoxemia [152]. Consequently, these two events promote an inflammatory-related pathway, which causes oxidative stress, negatively affecting mitochondrial functions and the production of MT [42]. However, MT can positively regulate its own production by increasing acetyl-CoA production from the pyruvate dehydrogenase complex, which activates the mitochondrial melatonergic pathway as a co-substrate [42], as shown in Figure 1.

## 2. Melatonin in the Gut–Liver Axis of ALD

Circadian disruptions make the intestines more susceptible to alcohol-induced tissue injuries [153]. Because several microbial changes and intestinal MT alterations occur in the progression of ALD, MT supplementation has the potential to mitigate alcohol-induced gut dysbiosis and liver injury. More specifically, MT has a profound impact in the context of alcohol-induced intestinal permeability in the gut–liver axis. In one study, alcohol that was metabolized with CYP2E1 with subsequent ROS byproducts enhanced the expression of circadian (i.e., CLOCK and PER2) proteins (loosely affiliated with MT), which were later shown to be involved in ethanol-induced intestinal hyperpermeability [154]. Circadian disruptions (perhaps due to abnormal MT levels) from irregular patterns of eating, sleeping, working, and drinking should be minimized for people who struggle with AUD or ALD [154]. Related to circadian genes, MT depletion and circadian disruption in human studies increased night shift workers’ susceptibility to intestinal permeability during social drinking via the upregulated transcription of *CLOCK* and *BMAL1* genes [155]. Depleted MT was also a sign of increased intestinal permeability in people with AUD who may suffer from sleep loss as a side effect of alcohol consumption [113]. Withdrawal from chronic alcohol consumption in people struggling with AUD was shown to increase MT levels because secretion of MT is regulated by the noradrenergic system, which is activated when withdrawing from alcohol [156].

MT supplementation has a unique set of roles in preventing alcohol-related disorders. First of all, MT exerts an anti-stressor effect which includes (a) antioxidant effects (i.e., neutralizing hydroxyl radicals, lipid peroxides, and peroxynitrite, and regulating the activities of glutathione peroxidase, superoxide dismutase, glucose 6-phosphate dehydrogenase (G6PDH), and nitric oxide synthase) and (b) anti-inflammatory effects to prevent aging [114]. MT also has been implicated in various signaling pathways such as the epidermal growth factor receptor–Brahma-related gene-1–telomerase reverse transcriptase (EGFR-BRG1-TERT) axis [157], Sirtuin1-Wnt/β-catenin-NOD-, LRR- and pyrin domain-containing protein 3 (SIRT1-Wnt/β-catenin-NLRP3) axis [158,159], SIRT1-Cereblon-Yin Yang 1-CYP2E1 (SIRT1-CRBN-YY1-CYP2E1) axis [160], and matrix metalloproteinase-9–nuclear factor–kappa B (MMP9-NF-κB) axis [161]. MT can also exhibit hepatoprotective effects when taken with other drugs, such as celecoxib (a nonsteroidal anti-inflammatory drug), by suppressing alcohol-induced apoptosis and inflammation via JNK and TNF-α signaling cascades and by mitigating oxidative stress [162]. MT also acts comparably to metformin [163] by upregulating the activity of AMPK and attenuating alcohol-related lipid accumulation to mitigate steatosis [164]. Specific to ALD and AUD, MT targets circadian rhythm disorders, alcohol withdrawal, and sleep apoplexy while acting as an antidepressant and sedative [22].

The preventive/therapeutic effects of MT on interconnected organ systems have been shown in many studies on ALD affecting the gut–liver axis [109,165]. MT has been shown to attenuate sub-clinical LPS and oxidative stress caused by alcohol [166] and to ameliorate alcohol-induced bile acid synthesis by enhancing miR-497 expression [167].

### Translational Research on the Effects of MT on the Progression of ALD

Many laboratories have reported the effects of MT on ALD in various types of animal studies (Table 1).

## 3. MT in the Gut–Liver Axis of MASLD

The kynurenine, indole, serotonin, and MT pathways are interconnected, and tryptophan metabolism is an integral driver in the progression of MASLD via the gut–liver axis and the immune system [168]. Related to the gut–liver axis, intestinal MT production levels, in sync with feeding cycle cues, may compensate for many factors of overnutrition accompanied with the progression of MASLD. For example, MT supplementation could potentially treat the obesity-related vicious cycle of insulin resistance, lipogenesis, and oxidative stress in the progression of MASLD/MASH by contributing to more brown adipose tissue (rich in mitochondria) in the body [169,170]. One study detailing the beneficial role of MT in managing obesity demonstrated that MT supplementation stimulated thermogenesis in brown fat cells and attenuated lipogenesis in white fat cells and hepatocytes to support that MT can prevent the development or progression of MASLD caused by a Western-style HFD [171]. MT may also regulate feeding hormones such as ghrelin [20] and the inflammatory processes [172,173,174] that commonly contribute to obesity-related MASLD [20].

The preventive and therapeutic effects of this antioxidant-like hormone have been shown to positively affect the gut–liver axis in multiple animal studies modeling MASLD [168,175,176,177]. MT has been shown to inhibit the metabolic effects of fructose [178], which have been found to contribute to MASLD [64,179]. MT has also been shown to alleviate gut-related liver inflammation from Ochratoxin A by restoring TJ/AJ proteins in the physical barrier of the gut, restoring liver function, and returning inflammatory markers connecting the gut and liver (i.e., LPS, interleukins, and TNF-α) [180]. Catalase plays a role in MASLD by preserving mitochondrial function, and MT prevents MASLD by supporting the effects of catalase [181]. Catalase-KO mice (CKO) exposed to HFD showed cellular lipid accumulation and decreased mitochondrial biogenesis, which were both treated with MT administration [181]. Overall, MT prevented fatty liver development and maintained the mitochondrial integrity of hepatocytes [181]. In another model of a short-term HFD, MT was able to reduce the effects of several pathological factors on the gut–liver axis and fat accumulation in the liver [176]. MT has also acted in the orphan nuclear receptor subfamily 4 group A member 1/DNA-dependent protein kinase catalytic subunit/tumor protein 53 (NR4A1/DNA-PKcs/p53) axis to reduce mitochondrial fission and promote mitophagy to prevent MASLD [142]. Some theorize that MT can even prevent HCC, which has been known to be irreversible [165].

### Translational Research on the Effects of MT in MASLD

Many laboratories have reported the effects of MT on the progression of MASLD in various animal models (Table 2).

## 4. Acute Toxicity in the Gut–Liver Axis: MT and Septic Hepatotoxicity

MT has been shown to affect septic hepatotoxicity, an acute liver condition, in similar ways to the way by which it prevents the beginning stages of ALD and MASLD. In one model of LPS-induced acute hepatotoxicity by cecal ligation and puncture, MT upregulated SIRT3 and SIRT1, which mitigated oxidative stress (by increased SOD2 activity via protein deacetylation), preserved mitochondrial function, and mitigated autophagy of intestinal epithelial cells. In this model, MT also attenuated inflammation by upregulating deacetylated nuclear factor-kappa B (NF-κB) and interleukin-10 (IL-10), and decreasing serum levels of TNF-α and interleukin-6 (IL-6) in the gut–liver axis [185]. In another septic hepatotoxicity study, MT alleviated LPS-induced hepatic SREBP-1c activation. It also mitigated lipid accumulation in mice which was shown via the significant decrease of liver weight, liver-to-body weight ratio, serum levels of very low density lipoproteins (VLDL), serum triglyceride (TG) levels, SREBP-1 expression, *fatty acid synthase* (FAS) mRNA expression, and *acetyl-CoA carboxylase* (ACC) mRNA expression, and increased *liver X receptor α (LXR-α)* expression [183]. These results suggest the beneficial role of MT as an alleviating compound in the beginning stages of endotoxin-evoked MASLD [183]. MT also mitigates cellular stress in the liver of septic mice by reducing ROS and increasing the unfolded protein response via the upregulation of PKR-like ER kinase (PERK) and C/EBP homologous protein (CHOP), and preventing the downregulation of cyclic AMP-response element-binding protein H (CREBH) [186].

## 5. Concluding Remarks

### 5.1. Suggestion of Behavioral Changes That May Increase MT to Prevent or Treat Liver Diseases

To efficiently manage ALD and MASLD, we should understand the underlying mechanisms of liver disease stages, including the bidirectional interactions of the gut–liver axis [88,150,187]. One example is that compensatory mechanisms occur to mitigate oxidative stress at the beginning stages of MASLD, but these compensatory mechanisms tend to fail at later, irreversible stages (i.e., cirrhosis) [88,187].

Exercise effectively prevents MASLD by mitigating oxidative stress and inflammatory factors, reducing intrahepatic fat content, increasing β-oxidation of fatty acids, overexpressing PPAR-γ, and attenuating apoptosis [188]. However, it is crucial to exercise at wakeful moments of the day because too much exercise during restful hours disrupts MT secretion [189], which does not allow for the prevention of liver diseases if MT is already low in biological stores.

Other lifestyle changes can be used to target oxidative stress to ameliorate liver and intestinal diseases, such as losing weight, eating beneficial diets with probiotics, and eliminating liver disease risk factors (e.g., overconsumption of alcohol, candy, and high-fructose corn syrup, and fast food-HFD) [70]. However, compliance with lifestyle changes could be improved. Increasing one’s antioxidant supply to maintain a homeostatic balance of ROS and antioxidants could be a promising approach and make patients more likely to stay compliant. Using antioxidants (like MT) that target mitochondria has a tremendous therapeutic potential to prevent and treat oxidative stress-related disorders because they cross the mitochondrial membranes [90,190].

Many dietary factors have been shown to prevent gut dysbiosis and LPS-induced ALD and MASLD that may be related to low MT production. Prebiotics (e.g., oligofructose, galactose, and inulin) and probiotics (e.g., *Bifidobacterium longum*, *Bifidobacterium infantis*, *Lactobacillus plantarum*, and *Lactobacillus acidenophilus*) [191] may have a role in upregulating MT production, possibly via bacterial L-tryptophan production (shown in a mouse model of ALD) [192]. Foods containing polyphenols (i.e., grape seed proanthocyanidins, resveratrol, quercetin, genistein, and isoflavones), other dietary compounds (i.e., rhein, phlorizin, capsaicin, rutin, lycopene, broccoli sprout extract, cranberry extract, and green tea extract), seeds (i.e., Alfalfa, almonds, anise, black mustard, celery, coriander, fennel, fenugreek, flax, green cardamom, milk thistle, poppy, sunflower, white mustard, and wolf berries), nuts (i.e., walnuts), fruits (i.e., apples, bananas, grapes, kiwis, pineapples, strawberries, tart cherries, and tomatoes), vegetables (i.e., asparagus, cabbage, carrots, celery, cucumbers, Indian spinach, Japanese radishes, beets, and corn), starches (i.e., oats, rice, tall fescue, and barley), and oil (i.e., olive oil) are all sources of MT, with seeds being the richest sources of MT [191,193,194]. In general, the Mediterranean Diet is a rich source of MT and has been demonstrated to prevent gut dysbiosis as well as intestinal and systemic inflammation [194,195].

### 5.2. Discussion of Future Studies and Considerations

Eating foods that contain MT may prevent the progression of liver diseases via the gut–liver axis, but if food consumption is not enough to raise serum MT levels then supplementation may be necessary [194]. The United States Food and Drug Administration does not regulate MT or supplement use [196]. In other countries, MT is prescribed and used for its intended purposes. In the United States, continuing the regulation of MT as a supplement may be part of the reason why 1 in 8 people taking MT take more than 5 milligrams (resulting in peak concentrations that last more than 4–8 h) and why it is being utilized for other benefits than its intended purpose of treating circadian rhythm disorders [197]. Supplementation should not be used for medical use until it is FDA-regulated, but if MT is administered then serum 6-hydroxymelatonin levels should be measured. A clinical trial in healthy men noted that in groups given 20 mg, 30 mg, 50 mg, or 100 mg MT, serum levels of the MT metabolite, 6-hydoxymelatonin sulfate, fluctuated less between participants in the same dosage groups than serum MT levels [198], reflecting that perhaps 6-hydroxymelatonin sulfate should be a biomarker of MT metabolism in the body.

Few clinical studies on MT supplementation in patients with ALD or MASLD have been detailed, especially from the perspective of the gut–liver axis, and this deserves more research [194,199]. However, MT has been found to be safe for people undergoing major liver surgery in doses of 50 mg/kg of body weight [200]. Since cirrhotic patients produce less MT [174] and may have high Model for End-Stage Liver Disease (MELD) scores that make them eligible for liver transplantation surgeries, MT supplementation may serve multiple purposes for cirrhotic patients [174,200]. Perhaps some of the animal studies detailed in Table 1 and Table 2 may be translated into clinical research study designs in order to promote this field of research.

One reason and consideration for designing MT supplementation clinical trials is that MT has been shown to be safe for supplementation in clinical studies. It can be taken in amounts up to 300 mg/day [157,198,201,202,203,204,205,206,207], with only reported mild side effects of dizziness, sleepiness, nausea, and headaches [204,208,209]. One case report detailed a female successfully using MT as her primary treatment for Multiple Sclerosis in a range of 50–300 mg daily for four years [210].

This brings us to the next consideration for further research, which is using MT as a complementary therapy for liver diseases. Co-administration with MT has been shown to increase the efficacy of drugs in treating and preventing the onset of liver diseases [211,212]. MT may also be taken in tandem with drugs that cause oxidative stress because it may allow higher doses of drugs to be used in cases where their usages are limited by toxicity [162,211]. MT has not been shown to interfere with drugs, but for future studies, for drug-drug interactions with compounds that are also metabolized by CYP1A2 [213], TPH, AADC, AA-NAT, or ASMT should be considered. Expanding our understanding of how MT co-administration may prevent liver diseases via the gut–liver axis in clinical trials is a promising direction for the future of liver disease research.

### 5.3. Conclusions

In this review, we summarized how MT may exert its benefits within the gut–liver axis to prevent the development and progression of ALD and MASLD. MT has many functions in mitigating oxidative stress and LPS-induced inflammation, preventing damage to the intestines and liver, and replenishing mitochondrial antioxidant levels. We also detailed many animal models in which MT supplementation prevents the progression of liver diseases and have discussed multiple exogenous sources of MT and lifestyle factors that affect human MT levels. Lastly, we considered aspects of MT supplementation in clinical trials, including its safety and information on the few known MT and liver disease clinical trials, with the hope of expanding the field of research in MT supplementation for the prevention and treatment of liver diseases via the gut–liver axis.

## Figures and Tables

**Figure 1 antioxidants-13-00043-f001:**
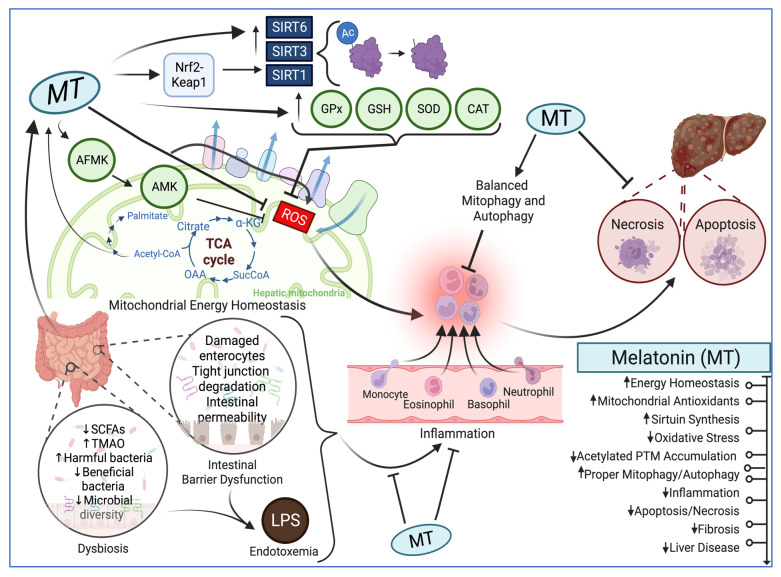
The beneficial effects of melatonin (MT) on the gut–liver axis and its underlying mechanisms. MT produced in the gut acts on hepatic mitochondria. Within hepatic mitochondria, MT can elevate the transcription factor Nuclear factor erythroid 2-related factor 2 (Nrf2) to increase the expression of its downstream antioxidant enzymes and Sirtuin isoforms, as illustrated. As an antioxidant, MT also prevents oxidative stress and promotes autophagy/mitophagy to activate proper mitochondrial metabolic functions and energy production. MT also attenuates gut dysbiosis and intestinal barrier dysfunction, inflammation, and endotoxemia to prevent liver disease via the gut–liver axis. The molecular effects of MT on the different systems are also listed. The up and down arrows indicate the increased and decreased cellular functions and systems, respectively.

**Table 1 antioxidants-13-00043-t001:** Effects of MT shown in animal models of ALD.

Stage of ALD and Model	Species Used	MT Dose and Treatment Duration	Signaling Pathway Affected by MT	Consequences of MT Effects	Reference
Acute Ethanol-Induced Stress	Male 2–3-month-old white mice	10 mg/kg for 10 days	↑ Mitochondrial function ↑ Mitochondrial respiration ↑ RCR, ADP/O, and V_ph_	Modulates oxidative phosphorylation; mitigates subclinical endotoxemia and oxidative stress	[166]
Steatosis/hepatitis	AML-12 cells and 7-week-old C57BL wild-type mice	10 μmol/L in cell model and 5 mg/kg for 10 days in Gao Binge Model	EGFR-BRG1-TERT pathway	Downstream effects of MT	[157]
Hepatotoxicity	Adult Male Rats	50 mg/kg for 11 consecutive days	↓ Serum transaminases ↓ ALP ↑ GSH, GST ↓ NO ↓ TNF-α, p-NF-κB, COX2 ↓ Hepatic cellular apoptosis	Decreased EtOH- induced apoptosis and inflammation via JNK and TNF-α signaling cascades	[162]
Steatosis	Mice and human samples	10 mg/kg orally for last 2 weeks of 4-week alcohol exposure Once daily for 7 days via tail-vein	CRBN-YY1-CYP2E1 ↓ CYP2E1 ↓ ROS ↓ Serum AST and ↓ALT ↓ IL-6 and ↓TNF-α ↓ Hepatic TGs ↓ Hepatic cholesterol	Induction of SIRT1 acts via the CRBN- YY1-CYP2E1 pathway to mitigate oxidative stress, improve liver function, prevent hepatic fat accumulation and inflammation	[160]
Fat accumulation	Adult male Sprague Dawley rats	20 and 40 mg/kg administration	↓ ALT, AST, and serum and hepatic TG ↑ SOD ↓ MDA ↑ p-AMPK ↑ MT1R expression	↑ AMPK ↓ Lipid accumulation	[164]
Steatosis	Female Balb/C Mice	15 mg/kg via i.p. prior to ethanol for 3 days	↓ MMP-9 activity, which then prevented NF-κB translocation to the nucleus after EtOH exposure ↓ Total pathology score but no significant effect on transaminases	Prevented inflammation	[161]
Chronic ALD	Mouse model	10 mg/kg daily oral gavage for last 2 weeks of 4 weeks of ethanol	↑ miR-497 expression	Ameliorates alcohol- induced bile acid synthesis by up- regulating miR-497 expression and attenuating the BTG2-YY1 pathway	[167]

Note: Upward arrows (↑) indicate increase or elevation, while downward arrows (↓) represent decrease or suppression of the specific parameter(s) after MT exposure(s).

**Table 2 antioxidants-13-00043-t002:** Effects of MT shown in animal models of MASLD.

Stage of MASLD or Liver Injury	Species Used in Study	MT Dose/ Treatment Duration	Signaling Pathway Affected by MT	MT Effects	Reference
Endotoxin- induced Hepatotoxicity	Female Wistar rats	10 mg/kg MT 30 min before LPS and 2 h after LPS	↓ LPS ↑ GSH levels ↑ SOD activity ↑ Catalase activity ↓ Serum nitrite NO_2_ ↓ TNF-α ↓ Hepatic necrosis	Mitigates endotoxin- induced hepatotoxicity ↑ Antioxidant stores ↓ Oxidative stress ↓ Hepatic inflammation and cellular death pathways	[182]
MASLD (Steatosis) ±HFD	Catalase-KO mice (CKO) and HepG2 cells	500 μg/kg/day MT for 6 weeks	↓ Liver weight ↓ Fat accumulation Restored Aspect Ratio (AR) and Form Factor (FF) values as measures of mitochondrial function ↑ mRNA expression of *FOXO1*, *PGC1β*, and *PPAR-γ* Improved mitochon- drial morphology ↑ mRNA expression of CPT1, CPT2, COX1, FGF21, Lcad, Mcad, Aconitase, IDH, SDH, MDH ↑ Hepatic [ATP]	Ablation of catalase plays a role in MASLD, and MT supports the function of mitochondria; HFD-exposed CKO mice exhibited cellular lipid accumulation and decreased mitochon- drial biogenesis, which was recovered with MT; MT prevented fatty liver development and main- tained mitochondrial in- tegrity in hepatocytes; Mitigated oxidative stress-induced mito- chondrial dysfunction and progression of MASLD	[181]
HFD-mediated MASLD lipogenesis and fibrosis	Liver-specific DNA PKcs-Knockout Mice NR4A1-Knockout mice	20 mg/kg/day MT via i.p. for 12 weeks after 12 weeks HFD or LFD	In HFD-fed mice, MT ↓ NR4A1 level ↓ Hepatocyte vacu- olization, steatosis, and fibrosis ↓ MMP9 ↓ VCAM1 ↓ IL-6 ↓ TNF-α ↓ TGF-β ↓ Mitochondrial ROS production ↓ Mitochondrial PTP opening ↑ ΔΨm mitochon- drial inner mem- brane potential	NR4A1/DNA-PKcs/p53 pathway, mitochondrial fission, and mitophagy; Prevented fat accumula- tion and fibrosis by inhibiting NR4A1. NR4A1 then activates Drp-1-mediated mito- chondrial fission, and repressing BNIP3-medi- ated mitophagy, which protects mitochondria; MT mitigated oxidative stress and calcium over- load by suppressing fission	[142]
MASH	Male 6–8-week-old CD-1 mice on a regular diet	5 mg/kg MT 30 min before 2 mg/kg LPS then 5 mg/kg 150 min after LPS	↓ LPS-induced acti- vation of SREBP-1c ↓ expression of *SREBP-1c* genes ↓ serum and hepatic TG levels	Prevents LPS-induced fat accumulation	[183]
Hepatitis	Male 1-day-old Cherry Valley ducklings	0.2 mg/mL MT supplemented in drinking water for 2 weeks	↓ *Bacteriodetes* induced by Ochratoxin A (OTA) ↑ *Firmicutes* ↓ *Bacteriodetes/Firmi-* *cutes* Ratio ↓ *Bacteroides* ↓ *Bacteroides uniformis* ↓ *Turicibacter sangui-* *nis*↓ Serum LPS levels ↑ Protein expression of Occludin and tight junction pro- tein-1 (TJP-1) ↑ Villi height and crypt depth ↑ Villi height/crypt depth ratio Restored gut histology ↓ TLR4, MyD88, p- -IKBα, p-IKBα/IKBα ratio, p-p65, liver IL-1β level, liver IL- 6, liver TNF-α ↑ Liver IL-10 ↓ Percentage of inflammatory liver cells in histology quantified	Anti-inflammatory Restored the physical barrier of gut Restored liver function and inflammatory markers	[184]
HFD-induced hepato-intestinal dysfunction and inflammation	Male Sprague Dawley rats	4 mg/kg/day for 2 weeks	↓ Perirenal fat ↓ Blood glucose ↑ TG levels by 3–4x ↓ Intestinal motility ↓ Liver weight ↑ GSH levels ↓ Myeloperoxidase (MPO) activity ↓ Serum ALT levels ↑ Hypertrophic goblet cell levels, epithelium, and brush border ↓ Vacuolization of hepatocytes Kept mitochondria intact	Restored Liver function and prevented fat accumulation Improved antioxidant levels and prevented oxidative stress Improved intestinal function and motility	[176]

Note: Upward arrows (↑) indicate increase or elevation, while downward arrows (↓) represent decrease or suppression of the specific parameter(s) after MT exposure(s).

## Data Availability

Figure 1 was created with software from Biorender.com.

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
