# Peer review of "Melatonin Prevents Alcohol- and Metabolic Dysfunction- Associated Steatotic Liver Disease by Mitigating Gut Dysbiosis, Intestinal Barrier Dysfunction, and Endotoxemia"

_antioxidants, 2023, doi:10.3390/antiox13010043_

Round 1

Reviewer 1 Report

Comments and Suggestions for Authors

The short review presents an interesting and important question of the role of melatonin in the prevention of alcohol-associated liver disease and metabolic dysfunction-associated steatotic liver disease and the effect of MT on the gut microbiome.

The review is somewhat chaotic, not optimally constructed, and needs re-edition.

For example, only half of the abstracts relates to the content of the review. The authors state the much higher production of melatonin in the gut than in the pineal gland but no numbers are given to document this statement. The reader may get an impression that CYP2E1 is localized only in the mitochondria, the microsomal enzyme is not mentioned at all.

 (Line 22: “ in quantities of 400-500 multitudes in”, unclear

Lines 23/24: “mechanistic functions of MT”

Lines 113-115. It is true that mitochondrial CYP2E1 can produce ROS but the main source of ROS in the cell is the one-electron leakage of the mitochondrial respiratory chain, which is worth mentioning

Line 128:  SCFA, please define on the first use

Line 139: “MT also prevents ROS”, unclear

Line 156: “may happen molecularly”, unclear

Line 218: Does lipid peroxidation produce hydroxyl radicals? Please provide citation(s)

Table 1: Which “Downstream effector of MT”?

Line 293: “no respiration”?

Line 322: “Thus”, the logical link of the second sentence with the first is not obvious

Line 352: “summarizes” is not appropriate in the Legend caption

Line 399 and next: Please write Latin species names in italics.

Please format the References according to the requirements of the journal.

Comments on the Quality of English Language

Some phrases not easy to understand, sometimes words are apparently lacking. Some words starting from capital letters in the middle of sentences.

Author Response

General Comments:
The short review presents an interesting and important question of the role of melatonin in the prevention of alcohol-associated liver disease and metabolic dysfunction-associated steatotic liver disease and the effect of MT on the gut microbiome. The review is somewhat chaotic, not optimally constructed, and needs re-edition. For example, only half of the abstracts relates to the content of the review. The authors state the much higher production of melatonin in the gut than in the pineal gland but no numbers are given to document this statement. The reader may get an impression that CYP2E1 is localized only in the mitochondria, the microsomal enzyme is not mentioned at all.
Response: We appreciate Reviewer 1 taking the time to read our manuscript and to provide specific and general comments on it. We have provided references next to our statement that the gut is responsible for producing 400-500 times more melatonin than the pineal gland. We also recognize the reviewer’s comment about Cyp2e1, and have added in statements that Cyp2e1 is mainly localized in microsomes but migrates to mitochondria, with references. Again, we thank Reviewer 1 for providing these thoughtful suggestions and have taken them seriously to improve our revised manuscript.

Specific Comment 1: Line 22: “ in quantities of 400-500 multitudes in”, unclear.

Response: We have changed this sentence to “is produced in 400-500 times greater concentrations in intestinal enterochromaffin cells”.

Specific Comment 2: Lines 23-24: “mechanistic functions of MT”.
Response: We have changed the phrase to “the functions of MT and molecular mechanisms by which it prevents”.

Specific Comment 3: Lines 113-115. It is true that mitochondrial CYP2E1 can produce ROS but the main source of ROS in the cell is the one-electron leakage of the mitochondrial respiratory chain, which is worth mentioning.
Response: We agree with the reviewer’s expert comment. Thus, we have changed the sentences (new lines 41-42 and 113-115 in the revised version) with the inclusion of an additional reference.

Specific Comment 4: Line 128: SCFA, please define on the first use.
Response: We have defined the full name of SCFAs (please see line 208 in the revised version).

Specific Comment 5: Line 139: “MT also prevents ROS”, unclear.
Response: For clarity, we have changed this sentence (please see line 288 in the revised version).

Specific Comment 6: Line 156: “may happen molecularly”, unclear
Response: We understand the comment, so we replaced this sentence.

Specific Comment 7: Line 218: Does lipid peroxidation produce hydroxyl radicals? Please provide citation(s).
Response: We appreciate the reviewer’s expert comment and thus have revised the sentence (please see line 315 in the revised version).

Specific Comment 8: Table 1: Which “Downstream effector of MT”?
Response: We thank the reviewer for catching up our mistake and have changed the word “effector” to “effects” to clarify an aspect of this referenced article in Table 1.

Specific Comment 9: Line 293: “no respiration”?
Response: We have changed “no respiration” to “low levels of respiration” (please see line 198 in the revised version).

Specific Comment 10: Line 322: “Thus”, the logical link of the second sentence with the first is not obvious.
Response: We appreciate the reviewer’s expert comment, so we changed the whole paragraph and removed the sentence (please see lines 342-355 in the revised version).

Specific Comment 11: Line 352: “summarizes” is not appropriate in the Legend caption
Response: That is a great point, so we deleted the word “summarizes” to accurately reflect that these are only some of all melatonin studies done.

Specific Comment 12: Line 399 and next: Please write Latin species names in italics.
Response: We agree with the reviewer’s suggestion and have changed all bacteria names and Latin-originated species names (please see line 206 and lines 424-425 in the revised version).

Specific Comment 13: Please format the References according to the requirements of the journal.
Response: We agree with the reviewer’s suggestion, so we downloaded the MDPI-modified ACS style for Endnote and have applied that to our manuscript.

Specific Comment 14: Comments on the Quality of English Language. Some phrases not easy to understand, sometimes words are apparently lacking. Some words starting from capital letters in the middle of sentences.
Response: Thank you for your expert comments and helpful suggestions. We have changed our text to make this manuscript grammatically correct, organized, fluid, logical, and structurally sound.

Reviewer 2 Report

Comments and Suggestions for Authors

Dear Authors,

I have read your paper with interest. It is an elegant descritption of potential role of melatonin in prevention/therapy of liver disease. Following issues should improve the paper in my opinion:

1. The list of human tissues in which melatonin is synthetized is longer than you have described it. Please, add some information about other sites of production of melatonin

2. You have not mentioned metabolites  of melatonin. Please , provide some examples of melatonin derivatives serving as anti-oxidant and anti-inflammatory processes.

3. Gut microbioma and blood-intestine barrier have a role in development of sepsis. Please, provide readers with some information about protective role of MT in sepsis.

4. You have listed translational studies on MT - unambigous information whether those studies were on animal models or with human patients woud be interesting

5. I would suggest adding some clinical information about using MT in patients with liver disease - if any is available.

Author Response

General Comments:
I have read your paper with interest. It is an elegant description of potential role of melatonin in prevention/therapy of liver disease. Following issues should improve the paper in my opinion:
Response: We greatly appreciate the reviewer’s time to review our article and provide overall and specific comments on our manuscript.

Specific Comment 1: The list of human tissues in which melatonin is synthetized is longer than you have described it. Please, add some information about other sites of production of melatonin.
Response: We greatly appreciate the reviewer’s expert comment. We changed that sentence in lines 51-67 of the revised version.

Specific Comment 2: You have not mentioned metabolites of melatonin. Please supply some examples of melatonin derivatives serving as antioxidant and anti-inflammatory processes.
Response: This is a great idea! Melatonin is incredibly unique in its ability to produce antioxidant metabolites as well. We added a paragraph on this information on new lines 227-228 of the revised version.

Specific Comment 3: Gut microbiome and blood-intestine barrier have a role in development of sepsis. Please, provide readers with some information about protective role of MT in sepsis.
Response: We appreciate the reviewer’s wonderful point. We wrote about how melatonin prevents septic hepatotoxicity in Section 4 and changed its title so that it stands out to readers. We welcome your further thoughts and suggestions in what we have mentioned there.

Specific Comment 4: You have listed translational studies on MT - unambiguous information whether those studies were on animal models or with human patients would be interesting.
Response: We understand the reviewer’s great point. We have revised our Tables to specifically mention the species assessed with melatonin.

Specific Comment 5: I would suggest adding some clinical information about using MT in patients with liver disease - if any is available.
Response: Thank you very much for this insight. There do not appear to be many randomized large-scale clinical studies regarding melatonin supplementation for liver diseases, especially through the gut-liver axis. However, there are studies that have been done to investigate the role of melatonin in other conditions, so we used these studies to described the safety and toxicity of melatonin (new lines 456-470 with new references 198-210 of the revised version). So, with this knowledge that melatonin is very safe and tolerable, we reconstructed the Conclusion to suggest that some of our animal studies in Tables 1 and 2 be adapted to clinical studies to advance this field of research. Overall, we added a few considerations when designing a clinical study, although melatonin is not regulated by U.S. FDA. Some clinical studies showed that high doses of MT have been well-tolerated in humans and effective in co-administration clinical trials, although some potential drug-drug interactions need to be considered as a precaution in future studies.

Reviewer 3 Report

Comments and Suggestions for Authors

In this paper, authors have reviewed the role of melatonin in alcohol- and metabolic dysfunction-associated steatotic liver disease. Paper is well written and comprehensive. I have just a remarks:

- table 1 lacks proper title

- an additional emphasis should be put into potential clinical perspectives and/or implications that can be concluded from this review

- also, clear reccomendation of still unknown, important characteristics related to this topic that should be investigated should be noted

Author Response

General Comments:
In this paper, authors have reviewed the role of melatonin in alcohol- and metabolic dysfunction-associated steatotic liver disease. Paper is well written and comprehensive. I have just a few remarks.
Response: Thank you for your wonderful suggestions that will allow our work to make a greater impact in this research area.

Specific Comment 1: Table 1 lacks proper title.
Response: We have now included a title for Table 1 or Table 2.

Specific Comment 2: An additional emphasis should be put into potential clinical perspectives and/or implications that can be concluded from this review.
Response: This is a wonderful suggestion. We added some clinical perspectives and implications, including our suggestions for behavioral modifications to consume foods rich in melatonin and to exercise in the morning so as not to affect melatonin production at night. Since melatonin supplementation is not regulated by the US FDA, we have also included considerations for melatonin supplementation, such as its apparent safety and beneficial effects on co-administration in clinical trials (please see our response above to the Comment #5 by Reviewer #2s).

Specific Comment 3: Also, clear recommendation of still unknown, important characteristics related to this topic that should be investigated should be noted.
Response: Thank you for this suggestion! We have recommended behavioral modifications that modulate melatonin and the gut-liver axis. In addition, we have described the evaluated safety of melatonin shown in several clinical trials to suggest that some of the animal studies listed in Tables 1 and 2 be adapted into clinical trials to advance this field of research. To elaborate on that suggestion, we included a few considerations for future clinical studies.

Reviewer 4 Report

Comments and Suggestions for Authors

This is a comprehensive review on MT being an antioxidant and its beneficial effects in treating ALD and MASLD in the context of understanding the gut-liver axis. The authors have provided an illustrative diagram and two tables summarizing the translational research of MT, which are very useful.

A few minor concerns are indicated below:

1) Giving a more detailed explanation of ALD and MASLD in the Introduction, rather than in later sections, is preferred.

2) Some proteins, eg. SIRTs, whose functions should be explained in the text.

3) Perhaps cases of MT overdose and adversary drug-drug interaction should be discussed as a precaution to avoid abusive (if any) use of MT. 

Author Response

General Comments:
This is a comprehensive review on MT being an antioxidant and its beneficial effects in treating ALD and MASLD in the context of understanding the gut-liver axis. The authors have provided an illustrative diagram and two tables summarizing the translational research of MT, which are very useful. A few minor concerns are indicated below.
Response: We greatly appreciate this reviewer’s time to review our article, and provide overall positive remarks and specific comments on our manuscript.

Specific Comment 1: Giving a more detailed explanation of ALD and MASLD in the Introduction, rather than in later sections, is preferred.
Response: This is a great suggestion! Thus, we rearranged the organization of this manuscript and now have dedicated sections 1.2 and 1.3 of the Introduction to detail the contributing factors of ALD and MASLD, respectively.

Specific Comment 2: Some proteins, e.g. SIRTs, whose functions should be explained in the text. Response: We agree with the reviewer’s comments. So, we have now included background information about the role of SIRTs and other proteins listed in the text.

Specific Comment 3: Perhaps cases of MT overdose and adversary drug-drug interaction should be discussed as a precaution to avoid abusive (if any) use of MT. 
Response: This is a great point. We have now included multiple statements in the Conclusion (Section 5) to address this point. There are few reports on overdoses of melatonin. Moreover, many clinical studies on melatonin conducted for other conditions have been reported to be safe in extreme amounts and beneficial when co-administered with toxic drugs. However, for the sake of the unknown, we have added a statement for readers who may want to advance this field of research to pay attention to any compounds that are metabolized by the same enzymes that also metabolize melatonin for potential drug-melatonin interactions.

Round 2

Reviewer 1 Report

Comments and Suggestions for Authors

I find the response and amendments satisfactory.